# QS-NeRV: Real-Time Quality-Scalable Decoding with Neural Representation for Videos

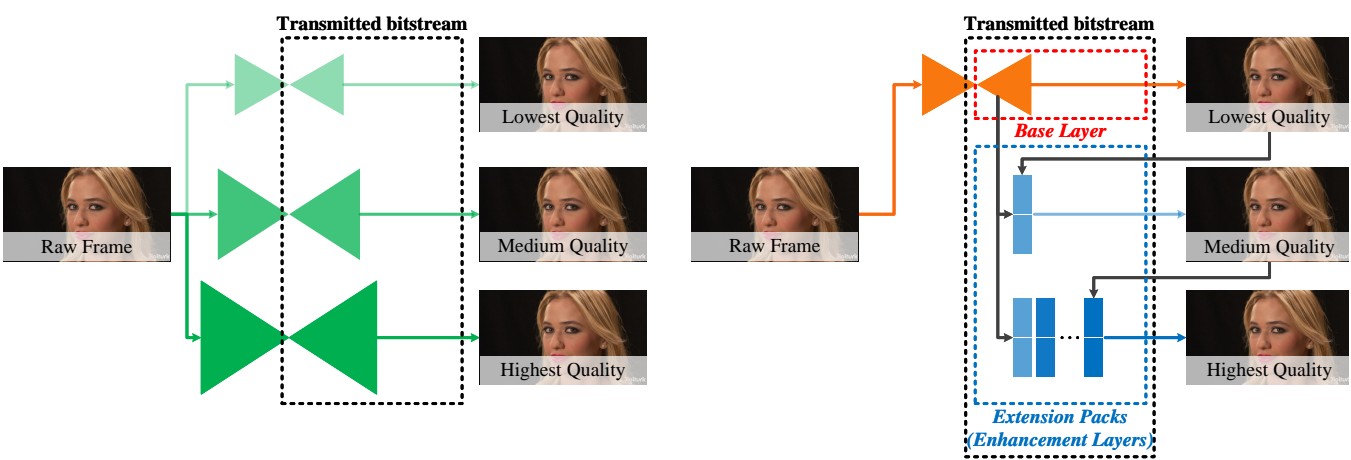

**Figure 1: (a) Existing INR-based methods typically train separate networks for each compression quality level of a video, which makes them difficult to apply to scalable compression. (b) Our proposed QS-NeRV can accommodate different quality requirements by integrating one base layer with one or more extension packs.**

## ABSTRACT

In this paper, we propose a neural representation for videos that enables real-time quality-scalable decoding, called QS-NeRV. QS-NeRV comprises a Self-Learning Distribution Mapping Network (SDMN) and Extensible Enhancement Networks (EENs). Firstly, SDMN functions as the base layer (BL) for scalable video coding, focusing on encoding videos of lower quality. Within SDMN, we employ a methodology that minimizes the bitstream overhead to achieve efficient information exchange between the encoder and decoder instead of direct transmission. Specifically, we utilize an invertible network to map the multi-scale information obtained from the encoder to a specific distribution. Subsequently, during the decoding process, this information is recovered from a randomly sampled latent variable to assist the decoder in achieving improved reconstruction performance. Secondly, EENs serve as the enhancement layers (ELs) and are trained in an overfitting manner to obtain robust restoration capability. By integrating the fixed BL bitstream with the parameters of EEN as an extension pack, the decoder can produce higher-quality enhanced videos. Furthermore, the scalability of the method allows for adjusting the number of combined packs to accommodate diverse quality requirements. Experimental results demonstrate our proposed QS-NeRV outperforms the state-of-the-art real-time decoding INR-based methods on various datasets for video compression and interpolation tasks.

## CCS CONCEPTS

- **Computing methodologies** → **Image compression**; *Reconstruction.*

## KEYWORDS

Video compression, Implicit neural representation, Quality-scalable, Real-time decoding

## 1 INTRODUCTION

Scalable video coding (SVC) constitutes a critical technology for adaptive video delivery, enabling dynamic, low-latency adjustments of video quality in alignment with varying network conditions, device capabilities, and user preferences. Especially within the realm of streaming media services, the implementation of real-time scalable decoding is vital for ensuring uninterrupted playback and enhance the quality of user experience. Its related standards, i.e., Scalable Video Coding (SVC) [25] and High Efficiency Video Coding Scalability Extension (SHVC) [2], have been developed. There are no less than two coding layers including one base layer (BL) and one or more enhancement layers (ELs) in standards. The BL

*ACM MM, 2024, Melbourne, Australia*

© 2024 Copyright held by the owner/author(s). Publication rights licensed to ACM.
ACM ISBN 978-x-xxxx-xxxx-x/YY/MM
https://doi.org/10.1145/nnnnnnn.nnnnnnn

contains essential information, while ELs provide additional details and higher video quality.

Recently, implicit neural representations (INR) have shown promising applications in video compression tasks. In contrast to learning-based explicit video compression methods [15, 17, 18], INR-based methods provide a simpler compression pipeline and accelerated decoding speed. INR-based methods transform video compression into a model compression problem by representing videos as implicit functions and encoding them into neural networks. NeRV [5] first proposes an image-based implicit representation method that utilizes frame indices as input and employs neural representation. HNeRV [4] improves NeRV into a hybrid structure by incorporating a trainable encoder that generates frame embeddings with more condensed information. DNeRV [34] additionally introduces frame differences to utilize explicit motion information. INR-based methods are characterized by slow encoding and real-time decoding, which gives them great potential for application in the field of video-on-demand (VoD).

However, it may be difficult for the existing INR-based methods to achieve quality scalability. There are two main reasons: (1) As shown in Fig. 1 (a), the encoding processes of videos at different quality levels are independent of each other. The compression of high-quality videos does not utilize or reference the low-quality videos, resulting in redundancy between their bitstreams. Additionally, the adoption of high-quality videos also involves discarding low-quality ones. (2) Achieving higher quality often requires larger models, and the drawback of prolonged training time associated with INR-based methods becomes more conspicuous as the scale of the models expands.

Inspired by the framework of SVC, we propose a neural representation for videos that enables real-time quality-scalable decoding, called QS-NeRV. QS-NeRV comprises a Self-Learning Distribution Mapping Network (SDMN) and Extensible Enhancement Networks (EENs). As shown in Fig. 1 (b), first, SDMN assumes the role of BL and is adopted to encode videos of the lowest quality. In SDMN, to realize efficient information transfer between the encoder and decoder for better reconstruction performance with lower bitstream overhead, we map the multi-scale information from the encoder to a specific distribution via invertible networks. Subsequently, during the decoding process, this information can be recovered from a randomly sampled latent variable and utilized to facilitate frame reconstruction. Secondly, EEN serves as an extension pack (also known as EL) and is trained in an overfitting manner to realize robust restoration capability. By exploiting the decoding information of the BL from the target frame and its adjacent frames, EEN of the current EL performs progressive quality enhancement on the output of the preceding layer. Higher-quality videos can be produced by combining the fixed BL bitstream with the parameters of EEN. To accommodate diverse quality requirements, the scalability of the method can be simply achieved by adjusting the number of combined EENs.

The main contributions are summarized as follows:

- We propose a Quality-Scalable NeRV, consisting of an SDMN and EENs, to achieve efficient scalable video encoding and real-time decoding. SDMN is the BL to obtain the lowest quality video, while EEN is the EL to provide higher-quality

video via referencing the preceding layer. QS-NeRV achieves adaptability to diverse quality requirements by integrating the BL bitstream with the parameters of EEN, while allowing for the adjustment of the number of packs.
- In SDMN, we establish an invertible skip connection between the encoder and decoder. Such distribution mapping and resampling approach enables efficient information transfer with lower bitstream overhead. Benefiting from the received auxiliary information, the decoder can achieve higher reconstruction performance.
- We evaluate our method on various datasets, and the experimental results demonstrate our proposed QS-NeRV is superior to the state-of-the-art real-time decoding INR methods on the video compression and interpolation tasks.

## 2 RELATED WORK

### 2.1 Scalable video compression

Scalable Video Compression focuses on the efficient compression of video data while accommodating various levels of quality, resolution, and bit rates. SVC [25] and SHVC [2], extends the capabilities of the previous video coding standard (H.264/AVC [30] and H.265/HEVC [26]) by introducing scalability features. They all employ a layered structure, consisting of a base layer (BL) and enhancement layers (ELs). BL contains essential information necessary for reconstructing a lower-quality version of the video, while ELs contain additional information that improves the quality and details of the video beyond the base layer. Decoders can selectively decode BL and one or more ELs based on available resources, network bandwidth, or user preferences. While deep neural networks (DNNs) have made great progress in image/video compression, there have been few studies that focus on the scalable compression task. [20] propose a learning-based end-to-end scalable compression model for images. [8] propose an interlayer restoration DNN (IRDNN) to improve the quality of the interlayer frame and coding efficiency of SHVC. There is an urgent need to fill the technology gap in the learning-based scalable video compression field.

### 2.2 Implicit neural representations

Implicit Neural Representations (INR) have emerged as a central paradigm that aims to parameterize signals (images, videos, 3D signals, etc.) as continuous functions. It maps input information, such as coordinate information, index information, and coded information, onto different types of signals to achieve a continuous representation of signals. INR has numerous applications, ranging from 3D object representation [6, 11] to view synthesis [1] and image/video reconstruction [23]. Recently, INR has shown great potential in the field of video compression [3, 5, 16, 19]. NeRV [5] first proposes a frame-indexed neural representation to achieve implicit video compression. Based on NeRV, CNeRV [3], E-NeRV [16], D-NeRV [7], and NIR VANA [19], which belong to the index-based NeRV methods, have achieved varying degrees of improvement. Nevertheless, this type of method does not provide enough content-specific coding information to the reconstruction network. HNeRV [4], which is a hybrid between implicit and explicit methods, achieves more competitive results by storing information in learnable frame-specific embeddings and a video-specific decoder.

Figure 2: The overall structure of our proposed QS-NeRV.

However, existing INR-based methods typically train separate networks for each compression quality level of a video, which results in unnecessary resource consumption when they are applied to scalable video compression.

### 2.3 Invertible neural networks

Invertible Neural Networks (INN) have received a considerable amount of attention since they were proposed by Dinh [10]. The INN is a two-shot structure that receives inputs from two parts and alternately couples the inputs through two or more neural networks. The INN is strictly efficiently reversible because of the reversibility of the coupling method. INNs are widely used in the fields of image hiding (HiNet [9], Steg-cINN [24], RIIS [33]), image compression [32], image rescaling (IRN [31], SelfC [27]), and image and video super-resolution [35]. In particular, IRN and SelfC learn INNs to map the high-frequency information of images to a specific distribution and then recover it from a randomly drawn latent variable, aiming to reduce the loss of high-frequency information caused by down-sampling operations during the image rescaling process. Inspired by this, we also adopt INNs to realize efficient information transfer between the encoder and decoder with lower bitstream overhead.

### 3 METHOD

### 3.1 Overview

The overall structure of our proposed Quality-Scalable NeRV (QS-NeRV) is depicted in Fig. 2. Our processing pipeline is conducted

in two main parts: base layer (BL) and enhancement layers (ELs). Firstly, the target frame $I_t$ is fed into a self-learning distribution mapping network (SDMN) to obtain the lowest quality output $O_t^{L_0}$ and the bitstream of BL $B_t^{L_0}$. Secondly, we train extensible enhancement networks (EENs) to serve as extension packs for further quality improvement. EENs are trained in an overfitting manner, where the lower-quality frame is utilized as the input, and its corresponding raw frame is employed as the label. Take EEN of the first EL as an example, the compression result of BL $O_t^{L_0}$ is fed into EEN for enhancement. By leveraging the decoding features of BL as auxiliary information, EEN can realize robust restoration capability and generate a higher-quality frame $O_t^{L_1}$. Similar to the combination of BL and one or more ELs in conventional scalable video coding, QS-NeRV achieves scalability by adjusting the number of EENs to accommodate diverse quality requirements.

### 3.2 Self-learning distribution mapping network

In SDMN, $I_t$ is first downsampled progressively by the encoder to extract features, and is eventually compressed into smaller frame embeddings $E_t^{L_0}$. We employ the encoder blocks of [4] to build our encoder. After receiving the frame embedding, the decoder reconstructs the decoded frame. We introduce a reparameterization-base decoder block, which improves upon the decoder block in [4]. As shown in Fig.3, during encoding (training), the HNeRV decoder Block uses only one convolution layer, while our decoder block can provide a more robust frame reconstruction capability by adopting

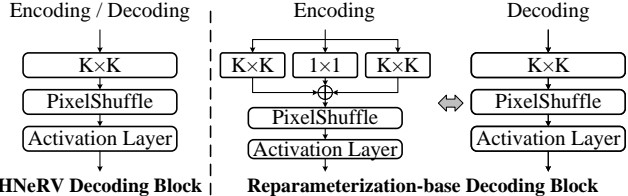

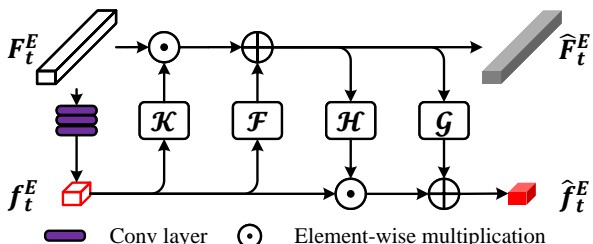

Figure 3: Illustration of the HNeRV decoder block (left) and the reparameterization-base decoder block (right). During encoding, our decoder block employs multiple branches, which can be merged into one convolution layer during decoding.

Figure 4: The structure of InvNet.

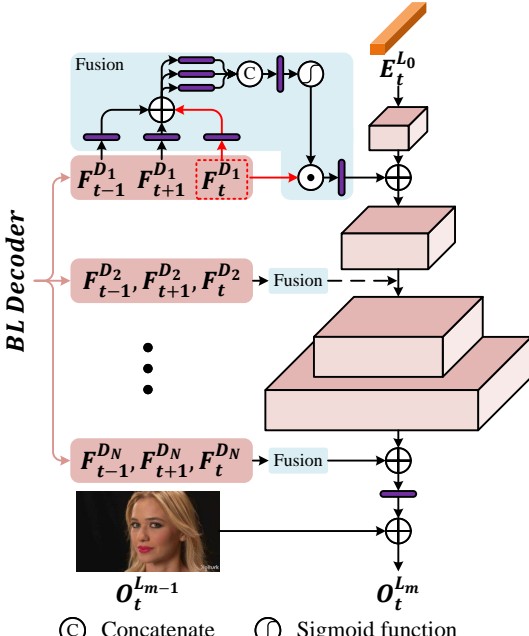

Figure 5: The structure of EEN.

multi-branch convolution layers of different sizes. During decoding, the learned kernels of multiple branches are merged into a single convolution kernel, thereby maintaining the same total number of parameters as the HNeRV decoder block. The decoder of BL is cascaded by $N$ reparameterization-base decoder blocks.

During the compression process, the inevitable information loss has an impact on the performance of the decoder. Establishing information transfer between the encoder and decoder can alleviate adverse effects, but directly transmitting whole features from encoder to decoder would result in a significant bitstream overhead. To address this problem, we adopt a distribution mapping and sampling approach by utilizing an invertible network (InvNet) to realize efficient information transfer between the encoder and decoder. As shown in Fig. 4, give the feature $F_t^E \in \mathbb{R}^{C \times H \times W}$ output by an encoder block, we first use multiple convolution layers to generate a compact latent $f_t^E \in \mathbb{R}^{1 \times H \times W}$ from $F_t^E$. Then, $F_t^E$ and $f_t^E$ are fed simultaneously into an InvNet for forward processing, which can be formulated as follows:

$$\hat{F}_t^E = F_t^E \odot \mathcal{K}(f_t^E) + \mathcal{F}(f_t^E)$$

$$\hat{f}_t^E = f_t^E \odot \mathcal{H}(\hat{F}_t^E) + \mathcal{G}(\hat{F}_t^E)$$

where $\mathcal{K}, \mathcal{F}, \mathcal{H}, \mathcal{G}$ are small DenseBlocks, and $\odot$ denotes element-wise product. In order to distinguish between different frames, we preserve the $\hat{f}_t^E$ as a frame-specific feature, while $\hat{F}_t^E$ is constrained to obey a Gaussian distribution $\hat{F}_t^E \sim \mathcal{N}(0, \sigma^2)$. Overall, the frame-specific feature $\hat{f}_t^E$ and the parameters of InvNet will be directly transmitted to the decoder as part of $B_t^{L_0}$.

During the decoding, $\hat{f}_t^E$ is received by the decoder, while $\hat{F}_t^E$ need only be sampled from a randomly drawn latent variable. The two features are fed into the transmitted InvNet for inverse processing. Benefiting from the strict and efficient reversibility of INNs, multi-scale information from the encoder can be recovered and used

to assist the decoder in achieving better reconstruction performance. For compressing a video containing $T$ frames, transmitting the entire feature incurs a significantly larger bitstream ($T \times C \times H \times W$) compared to our adopted approach ($T \times 1 \times H \times W + Param_{InvNet}$).

To summarize, $B_t^{L_0}$ contains multiple components, including the parameters of $E_t^{L_0}$, $\hat{f}_t^E$, $Decoder$, and $InvNets$.

$$B_t^{L_0} = E_t^{L_0} + \hat{f}_t^E + Decoder + InvNets$$

### 3.3 Extensible Enhancement Network

In order to attain scalability, we introduce an extensible enhancement network (EEN) as the enhancement layer. Inspired by SVC, whose encoding process of ELs is usually built upon the BL by encoding the differences to achieve progressive enhancement, we take the lower-quality frames, the frame embedding of BL $E_t^{L_0}$, and the features output by each decoding block of BL $F_t^{D_n}$ as inputs to train EEN in an overfitting way, where $n \in [1, N]$ is the block index. Considering that multi-frame reference facilitates the performance of the reconstruction, we augment EEN by incorporating the features of the neighboring frames, $F_{t-1}^{D_n}$ and $F_{t+1}^{D_n}$, as side information. As shown in Fig. 5, EEN also adopts a pyramid structure formed by stacking the reparameterization-base decoder blocks. In the beginning, $E_t^{BL}$ is fed into the first block for processing. Meanwhile, we fuse inter-frame information together. Firstly, we utilize $3 \times 3$ convolution layers to reduce the channels of the features generated by the first decoding block of BL to 1. Then, we perform element-wise summation to combine these squeezed features.

$$h_1 = C_{3 \times 3}(F_{t-1}^{D_1}) + C_{3 \times 3}(F_t^{D_1}) + C_{3 \times 3}(F_{t+1}^{D_1}),$$

where $C_{k \times k}$ denotes $k \times k$ convolution layer. After obtaining $h_1$, three convolution layers of different sizes are adopted to extract

**Table 1: The parameter quantities of QS-NeRV of different sizes over 1920×960 sequences containing 600 frames.**

| Models | Parameters | |
|---|---|---|
| | BL | ELs |
| QS-NeRV-3M | $E_t^{L_0}$: 0.077M $\hat{f}_t^E$: 0.182M | - |
| QS-NeRV-4.5M | *Decoder*: 2.800M *InvNets*: 0.182M | $B_t^{L_1}$ : 1.5M |
| QS-NeRV-6M | $B_t^{L_0}$ (*Total*): 3.251M | $B_t^{L_1} + B_t^{L_2}$ : 3M |

multi-scale information, and we integrate all information using a concatenation operation followed by a convolution layer. A spatial-wise attention map $M$ is generated by applying the sigmoid operation on the integrated result.

$$M = Sigmoid(C_{3\times3}([C_{1\times1}(h_1), C_{3\times3}(h_1), C_{5\times5}(h_1)])),$$

where $[\cdot, \cdot]$ represents concatenation operation. Finally, we multiply $M$ by $F_t^{D_1}$ to get the fused result, which will be combined with the output of the first block as the input to the next block.

We use the same method to fuse the remaining BL features and apply them to progressive refinement. In the end, a residual image output by the last block will be added to the lower-quality frame to obtain a higher-quality one.

$$O_t^{L_m} = EEN_m(F_{t-1}^{D_n}, F_t^{D_n}, F_{t+1}^{D_n}) + O_t^{L_{m-1}},$$

where $EEM_m, m \in [1, M]$ denotes the $m$-th enhancement layer. The parameters of EENs will be used as the bitstream of ELs $B_t^{L_m}$ for transmission and storage. Benefiting from the combination of BL and ELs, we can adaptively adjust the number of adopted EENs based on quality requirements. Table 1 presents an example of parameter quantities for QS-NeRV of different sizes over 1920×960 sequences containing 600 frames.

# 4 EXPERIMENTS

## 4.1 Dataset

We validate the effectiveness of our proposed QS-NeRV on *Bunny* [4], UVG [21], and DAVIS datasets [29]. *Bunny* owns 132 frames at 1280×720. UVG comprises 7 videos with a resolution of 1920×1080 and lengths of either 600 or 300 frames. DAVIS consists of a group of 50 high-quality, 1920 × 1080 video sequences. These sequences exhibit diverse video compression challenges, including occlusion, motion blur, and appearance changes. We select *Bunny*, 7 videos from UVG, and 22 videos from DAVIS for evaluation. Then, we center-crop the videos into 1280 × 640 for *Bunny*, and 1920 × 960 for UVG and DAVIS as in [5] and [4].

## 4.2 Training Details

For SDMN, we firstly use the Mean Square Error (MSE) loss to minimize the difference between the output video and the original video. Moreover, we introduce a distribution loss to make sure that $\hat{f}_t^E$ is mapped to the correct distribution. The loss function for

**Table 2: PSNR on *Bunny* with different model sizes.**

| Real-time Methods | Size | | | Avg. |
|---|---|---|---|---|
| | 0.75M | 1.5M | 3M | |
| NeRV | 28.46 | 30.87 | 33.21 | 30.85 |
| E-NeRV | 30.95 | 32.09 | 36.72 | 33.25 |
| HNeRV | 32.81 | 35.57 | 37.43 | 35.27 |
| DNeRV | 32.39 | 35.21 | 37.82 | 35.14 |
| FFNeRV | 30.37 | 33.83 | 37.01 | 33.74 |
| **QS-NeRV** | **33.74** | **36.37** | **38.59** | **36.23** |

SDMN $\mathcal{L}_{SDMN}$ are defined as :

$$\mathcal{L}_{SDMN} = \|I_t, O_t^{L_0}\|_2 + (-\mathbb{E}_{q(I_t)}[\log p(\hat{F}_t^E)])$$

where N is the number of the training samples.

For EENs, we use the MSE loss for training networks, and the loss function for EENs $\mathcal{L}_{EEN}$ is defined as:

$$\mathcal{L}_{EEN} = \|I_t, O_t^{L_m}\|_2$$

We adopt Adam [12] as the optimizer, and the batch size is set as 1. The initial learning rate is set as $5e - 4$ with a cosine learning rate schedule. The training of both SDMN and the first EEN terminates after 300 epochs, while the latter EENs only need to be finetuned 150 epochs on top of the previously trained EEN. The stride list and kernel size settings of the decoding block in SDMN and EENs are kept consistent with [4]. We use PSNR and SSIM to measure the objective quality of the reconstructed video for our proposed QS-NeRV and the state-of-the-art real-time decoding model, including NeRV [5], E-NeRV [16], HNeRV [4], DNeRV [34] and FFNeRV [14]. All experiments are conducted in PyTorch [22] with GPU RTX4090.

## 4.3 Video Regression

The results of NeRV [5], E-NeRV [16], HNeRV [4] and FFNeRV [14] are consistent with their original paper. To ensure a fair comparison, we modify the model of DNeRV [34] to meet the predetermined size requirements and retrain it on our dataset.

**Bunny**. We select SDMN with 0.75M parameters as BL. To extend the QS-NeRV to versions with 1.5M and 3M parameters, the parameters of EENs of the first and second ELs are set to 0.75M and 1.5M, respectively. As shown in Table 2, the comparison results illustrate that QS-NeRV outperforms other methods across different sizes.

**UVG**. For UVG videos with resolutions of 1920×960, the model size of BL is set to 3M parameters. Then, we scale up the model parameters to 4.5M and 6M by adding EENs with 1.5M parameters. The results are reported in Table 3. We further implement performance validation on the downsampled version of UVG (960 × 480), and the results are shown in Table 4. Benefiting from the establishment of information flow between the encoder and decoder in SDMN, the performance of BL has been dramatically improved compared with other methods. Combining with BL and leading performance on multiple videos also proves the effectiveness of our EENs. Fig. 6 shows that the reconstruction performance of QS-NeRV is superior, as it can generate richer details.

**DAVIS**. As for DAVIS, we compare the performance of models with 3M parameters. As shown in the Table. 5, the results of ten sequences demonstrate that our proposed QS-NeRV achieves the most significant boost in terms of PSNR and SSIM.

**Table 3: PSNR (dB) on UVG at resolution 1920×960. Bold means best results.**

| Real-time Methods | Size | Beauty | Bosph | Bee | Jockey | Ready | Shake | Yacht | Avg. |
|---|---|---|---|---|---|---|---|---|---|
| NeRV | 3M | 33.25 | 33.22 | 37.26 | 31.74 | 24.84 | 33.08 | 28.03 | 31.63 |
| E-NeRV | | 33.17 | 33.69 | 37.63 | 31.63 | 25.24 | 34.39 | 28.42 | 32.02 |
| HNeRV | | 33.58 | 34.73 | 38.96 | 32.04 | 25.74 | 34.57 | 29.26 | 32.69 |
| DNeRV | | 33.67 | 34.79 | 39.22 | 32.58 | 26.24 | 34.71 | 28.80 | 32.86 |
| FFNeRV | | 33.57 | 35.03 | 38.95 | 31.57 | 25.92 | 34.41 | 28.99 | 32.63 |
| **QS-NeRV (BL)** | | **33.92** | **35.74** | **39.27** | **33.83** | **27.67** | **35.04** | **29.97** | **33.63** |
| HNeRV | 4.5M | 33.89 | 35.48 | **39.43** | 32.90 | 26.73 | 35.17 | 30.01 | 33.37 |
| DNeRV | | 33.80 | 35.06 | 39.30 | 33.60 | 27.36 | 34.99 | 29.60 | 33.39 |
| FFNeRV | | 33.78 | 35.37 | 39.41 | 34.13 | 27.45 | 34.70 | 29.34 | 33.45 |
| **QS-NeRV (BL+EL×1)** | | **34.11** | **36.69** | 39.38 | **34.89** | **28.75** | **35.68** | **30.94** | **34.35** |
| HNeRV | 6M | 34.03 | 36.04 | 39.51 | 33.65 | 27.44 | 35.89 | 30.72 | 33.90 |
| DNeRV | | 33.84 | 35.35 | 39.30 | 34.07 | 27.78 | 35.09 | 29.51 | 33.56 |
| FFNeRV | | 33.98 | 36.63 | **39.58** | 33.58 | 27.39 | 35.91 | 30.51 | 33.94 |
| **QS-NeRV (BL+EL×2)** | | **34.21** | **37.19** | 39.42 | **35.48** | **29.39** | **36.07** | **31.57** | **34.76** |

**Table 4: PSNR (dB) on UVG at resolution 960×480. Bold means best results.**

| Real-time Methods | Size | Beauty | Bosph | Bee | Jockey | Ready | Shake | Yacht | Avg. |
|---|---|---|---|---|---|---|---|---|---|
| NeRV | 3M | 36.27 | 35.07 | 40.76 | 32.58 | 25.81 | 35.33 | 30.11 | 33.70 |
| E-NeRV | | 36.26 | 36.06 | **43.26** | 32.70 | 26.19 | 35.64 | 30.38 | 34.35 |
| HNeRV | | **36.91** | 36.95 | 42.05 | 33.33 | 27.07 | 36.97 | 30.96 | 34.89 |
| DNeRV | | 32.78 | 34.53 | 38.52 | 32.22 | 25.94 | 34.23 | 29.07 | 32.47 |
| FFNeRV | | 35.27 | 34.84 | 41.46 | 33.13 | 26.27 | 35.02 | 29.14 | 33.59 |
| **QS-NeRV (BL)** | | 35.15 | **37.32** | 40.73 | **35.00** | **29.02** | **37.02** | **32.38** | **35.06** |
| HNeRV | 4.5M | **37.27** | 37.81 | **42.23** | 34.76 | 27.45 | 37.38 | 32.09 | 35.57 |
| DNeRV | | 32.97 | 35.35 | 38.68 | 33.43 | 27.30 | 34.70 | 29.80 | 33.18 |
| FFNeRV | | 35.76 | 36.41 | 42.06 | 34.62 | 27.89 | 35.88 | 30.44 | 34.72 |
| **QS-NeRV (BL+EL×1)** | | 35.49 | **38.76** | 40.96 | **36.56** | **30.84** | **37.58** | **34.30** | **36.17** |

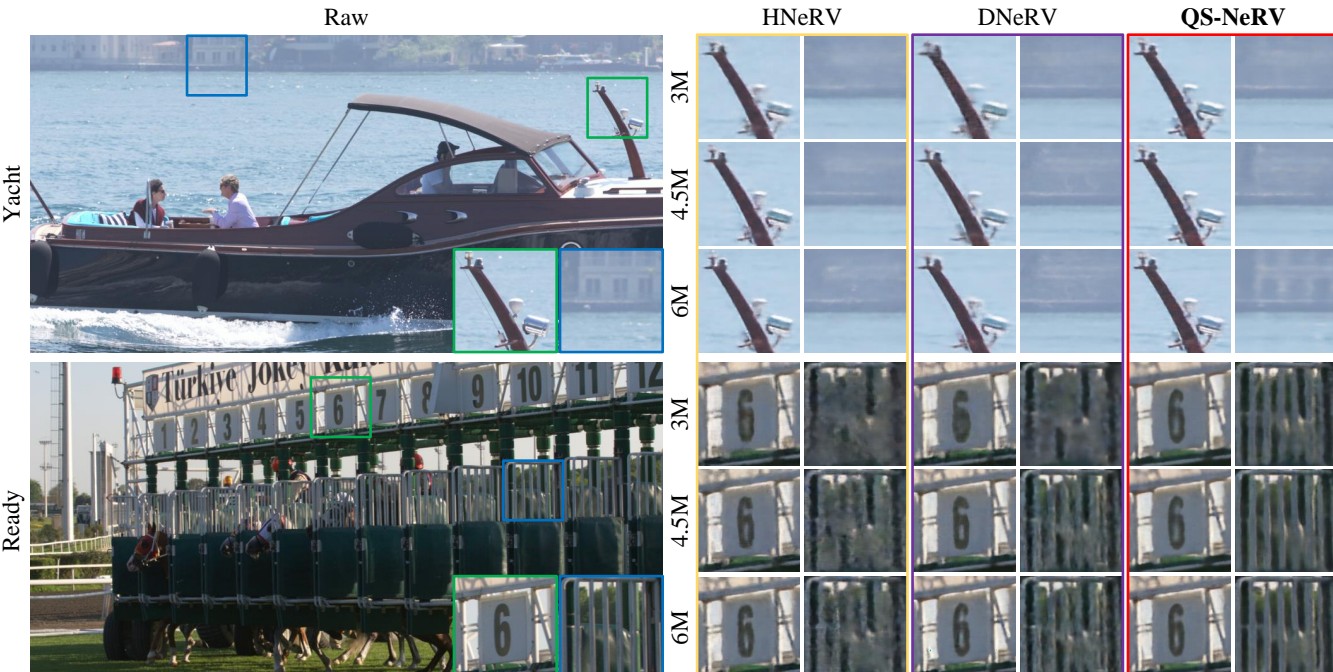

**Figure 6: Visualization example of video neural representations at various model sizes.**

## 4.4 Downstream Tasks

*4.4.1 Scalable Video Compression.* We compare QS-NeRV with SHM 12.4 [2], X264 [30], X265 [26], multiple real-time decoding INR-based methods [4, 5, 14, 34] and a learning-based video compression method, i.e., DCVC [15]. We quantize the INR-based models in 8 bits with entropy encoding and without model pruning. To present more

intuitive, rate–distortion curves of our and other methods over the UVG dataset are shown in Fig. 8. In this figure, we can observe that, except for the official coding tool SHM 12.4 and DCVC, the curve of QS-NeRV is above that of others. QS-NeRV surpasses traditional video codecs H.264 or H.265 in PSNR, and it is also superior to the state-of-the-art INR-based methods. Although learning-based video

Table 5: PSNR (dB) and SSIM on DAVIS at resolution 1920×960. Bold means best results.

| Video | Video Regression | | | | Video Interpolation | | |
|---|---|---|---|---|---|---|---|
| | HNeRV | DNeRV | FFNeRV | QS-NeRV | DNeRV | FFNeRV | QS-NeRV |
| Blackswan | 30.35/0.891 | 30.20/0.898 | 30.67/0.940 | **32.78/0.942** | 27.40/0.834 | 26.30/0.786 | **30.31/0.902** |
| Bmx-trees | 28.76/0.861 | 28.64/0.858 | 29.06/0.912 | **31.08/0.917** | 24.95/0.710 | 25.18/0.725 | **25.65/0.755** |
| Car-shadow | 31.32/0.936 | 30.23/0.924 | 33.06/**0.964** | **33.89**/0.954 | 26.29/0.872 | 27.63/0.883 | **29.70/0.927** |
| Cows | 24.11/0.792 | 24.29/0.798 | 22.36/0.707 | **25.13/0.842** | 23.06/0.752 | 22.36/0.707 | **24.36/0.821** |
| Dog | 30.96/0.898 | 31.10/0.900 | 31.22/0.936 | **33.35/0.943** | 26.36/0.724 | 26.95/0.757 | **28.69/0.798** |
| Drift-straight | 30.80/0.932 | 30.56/0.928 | 31.29/0.962 | **34.24/0.968** | 23.72/0.726 | 24.90/0.730 | **25.63/0.738** |
| Goat | 26.62/0.858 | 26.90/0.863 | 25.62/0.874 | **28.61/0.906** | 22.27/0.634 | 22.06/0.650 | **24.65/0.761** |
| Mallard-fly | 29.22/0.848 | 28.98/0.839 | 29.89/**0.915** | **30.88**/0.897 | 24.58/0.684 | 25.14/**0.724** | **25.51**/0.711 |
| Parkour | 26.56/0.851 | 26.69/0.853 | 26.97/0.874 | **28.01/0.891** | 23.31/0.739 | 23.85/0.774 | **24.55/0.791** |
| Scooter-black | 27.38/0.923 | 27.86/0.932 | 27.75/0.954 | **31.26/0.961** | 20.69/0.708 | 21.90/0.719 | **22.92/0.750** |
| Average | 28.61/0.879 | 28.55/0.879 | 28.79/0.904 | **30.90/0.922** | 24.26/0.743 | 24.63/0.745 | **26.20/0.795** |

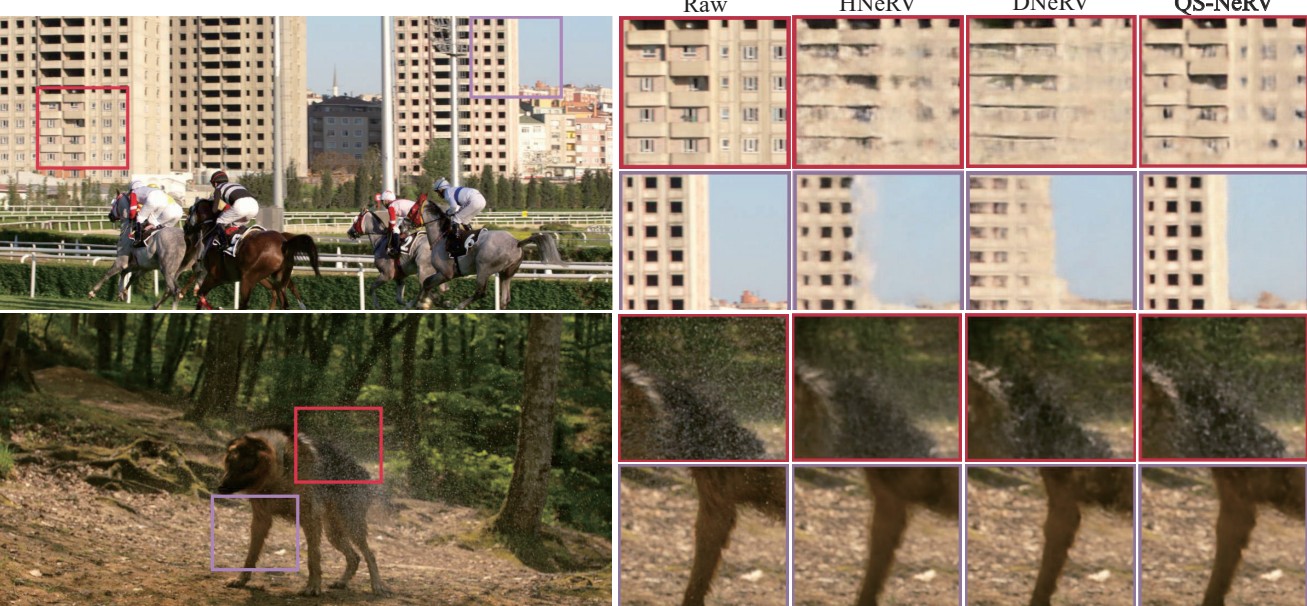

Figure 7: Visualization example of video interpolation at 3M model size.

Table 6: Video interpolation results on 1920 × 960 UVG in PSNR. Bold means best results.

| Video | Beauty | Bosph | Bee | Jockey | Ready | Shake | Yacht | Avg. |
|---|---|---|---|---|---|---|---|---|
| HNeRV | 31.10 | 34.38 | 38.83 | 23.82 | 20.99 | 32.61 | 27.24 | 29.85 |
| DNeRV | 30.58 | 34.71 | 39.02 | 23.50 | 20.07 | 32.50 | 27.09 | 29.64 |
| FFNeRV | 30.43 | 32.77 | 38.58 | **26.84** | 23.18 | 31.78 | 26.31 | 29.98 |
| QS-NeRV | **32.85** | **34.80** | **39.17** | 25.45 | **24.21** | **34.27** | **29.11** | **31.38** |

compression methods exhibit superior compression efficiency, the inability to facilitate real-time decoding constitutes a major barrier to their practical application.

*4.4.2 Video Interpolation.* We assess the generalization capabilities of various methods through video interpolation tasks. Following the experimental setup outlined in [34], we present the quantitative results on UVG and DAVIS datasets in Table 6 and Table 5, respectively. For the UVG dataset, our method exhibited an average improvement of 1.40−1.53dB in PSNR compared to other methods. In terms of the DAVIS dataset, our performance of PSNR and SSIM on all videos exceeds the state-of-the-art method. Specifically,

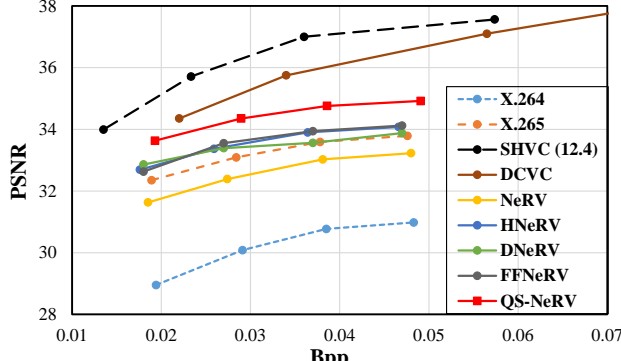

Figure 8: Compression results of scalable video compression on 1920 × 960 UVG.

comparing with the SOTA method, we obtained the largest improvement on sequence *Blackswan*, with an enhancement of 2.91dB. The maximum improvement in SSIM (0.111) is observed in sequence *Goat*. The qualitative results are presented in Fig. 7.

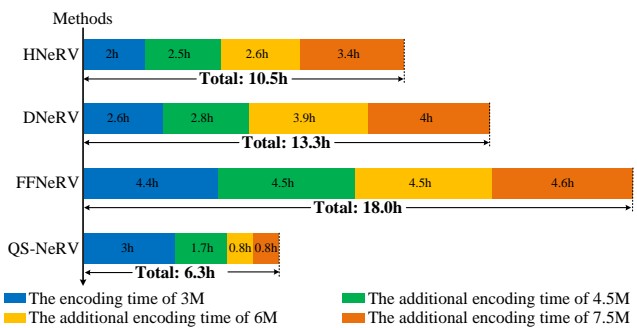

Figure 9: The speed comparison of scalable video encoding.

Table 7: The speed comparison of scalable video decoding.

| Methods | Param | Dec time (ms) ↓ | FPS ↑ | Real-time |
|---------|-------|-----------------|-------|-----------|
| HNeRV | | 8.1 | 124 | ✓ |
| DNeRV | | 7.9 | 127 | ✓ |
| FFNeRV | 3M | 14.1 | 71 | ✓ |
| HiNeRV | | 65.3 | 15 | ✗ |
| QS-NeRV | | 12.3 | 81 | ✓ |
| HNeRV | | 15.3 | 65 | ✓ |
| DNeRV | | 13.6 | 73 | ✓ |
| FFNeRV | 4.5M | 14.1 | 71 | ✓ |
| HiNeRV | | 77.4 | 13 | ✗ |
| QS-NeRV | | 12.3 + 9.2 | 47 | ✓ |
| HNeRV | | 18.0 | 55 | ✓ |
| DNeRV | | 14.9 | 67 | ✓ |
| FFNeRV | 6M | 14.1 | 71 | ✓ |
| HiNeRV | | 90.0 | 11 | ✗ |
| QS-NeRV | | 12.3 + 9.2 + 9.2 | 33 | ✓ |

## 4.5 Processing Speed

**Encoding time**. Here, we analyze the scalable video encoding time, which is equivalent to the training time of the network for the INR-based methods. The evaluation is conducted on the video containing 600 frames of 1920×960. As shown in Fig. 9, the encoding time of SDMN (3M) is 3 hours, which is slightly longer than [4] (2 hours) and [34] (2.6 hours). However, when aiming to achieve higher-quality videos from existing lower-quality videos, other methods typically train a larger network from scratch, and their encoding time increases substantially. In contrast, our QS-NeRV only needs to train smaller networks, i.e., EENs, as extension packs. Moreover, from the second EL onwards, EEN can be finetuned by fewer epochs on top of the previously trained EEN. In brief, as the demand for video quality increases, the advantages of QS-NeRV in scalable video encoding become increasingly apparent.

**Decoding time**. We further test the decoding time of QS-NeRV (Float32). As shown in Table 7, QS-NeRV is capable of delivering 1920×960 video at various quality levels in real-time (>30 fps). To satisfy the demand for different qualities, other methods need to transmit all models to the decoding side for switching (3 + 4.5 + 6M), while QS-NeRV only needs to transmit the largest model (3 + 1.5 + 1.5M) and adjust the number of ELs, which implies that our decoding process is more flexible. It is worth mentioning that INR-based methods are characterized by slow encoding and real-time decoding, which gives them great potential for application in the field of video-on-demand (VoD). However, HiNeRV [13] sacrifices decoding speed to significantly improve performance, making it difficult to apply in the real world.

Table 8: The comparison of different skip connections, *UVG*.

| Type | Params (M) | | | PSNR (dB) |
|------|------------|------------|-------|-----------|
| | *Decoder* | Connection | Total | |
| w/o | 3.16 | 0 | 3.16 | 33.30 |
| Direct | 2.80 | 12.29 | 15.09 | 33.88 |
| Not invertible | 2.80 | 0.36 | 3.16 | 33.52 |
| Invertible | 2.80 | 0.36 | 3.16 | 33.63 |

Table 9: The comparison of different training strategies, *UVG*.

| Size | Training strategy | PSNR (dB) |
|------|-------------------|-----------|
| 4.5M | Joint | 34.19 |
| | Separate (3+1.5) | 34.35 |
| 6M | Joint | 34.48 |
| | Separate (3+1.5+1.5) | 34.76 |

## 4.6 Ablation Study

**The effectiveness of the invertible skip connection**. To verify the effectiveness of our proposed invertible skip connection, we compare its performance with no skip connection, direct skip connection, and not invertible connection. As depicted in Table 8, the advantages of establishing information transmission between the encoder and decoder are evident, resulting in remarkable improvement. Although achieving optimal performance by direct skip connection, the large data volume imposes a significant burden on the transmission. Our proposed method greatly reduces the transmitted data and a slight degradation in performance is acceptable. With the same number of parameters, we further rescale features to be transmitted in a not invertible way, e.g. using convolution layers. Comparative results show that information loss caused by not invertible connections leads to performance degradation.

**Joint vs. separate training**. There are two strategies to train our large model: one is to jointly train the BL and ELs, while the other is to train ELs after BL has been trained. On the one hand, [28] has proved that gradually fitting a distribution through multiple steps is preferable to fitting it in one step. On the other hand, EEN takes the output of the previous layer as input and learns a residual image, the learning process becomes simpler, leading to a greater enhancement. As shown in Table 9, separate training is better than joint training. Furthermore, separate training is the key to achieving scalable video coding.

## 5 CONCLUSION

In this paper, we achieve real-time quality-scalable decoding with an neural representation for videos (QS-NeRV). To realize scalability, we train an SDMN to be the base layer and EENs to be enhancement layers. Higher-quality videos can be produced by combining the BL bitstream with the parameters of EEN. By adjusting the number of combined EENs, QS-NeRV can accommodate diverse quality requirements. Moreover, we establish efficient information transfer between the encoder and decoder of SDMN by exploiting the distribution mapping and resampling approach, resulting in enhancing the reconstruction ability with lower bitstream overhead. Comprehensive comparisons across various aspects have consistently demonstrated the superior performance of our method compared to alternative methods.

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
