# OpenReview forum: "QS-NeRV: Real-Time Quality-Scalable Decoding with Neural Representation for Videos"
_acmmm.org/ACMMM/2024/Conference — MM2024 Poster_

### Official Review · Reviewer_xux1 · 2024-05-08

**Rating:** 3
**Confidence:** 2

**Summary:**

Video compression methods based on Implicit Neural Representations (INR) are challenging to apply to multi-bitrate video transmission due to the independent encoding of each quality level and the redundancy in transmission. Therefore, this paper combines NeRV with Scalable Video Coding (SVC) and proposes QS-NeRV, which employs the INR-based network SDMN to generate  the base layer and designs an Extensible Enhancement Network (EEN) to generate enhancement layers. Furthermore, this paper introduces Invertible Neural Networks (INN) to compress features to reduce bandwidth consumption.

**Strengths:**

1. The research motivation of this paper is clear, and the research approach is reasonable. The description is also very clear, making it concise and easy to understand.
2. Compared with other methods derived from NeRV, QS-NeRV is capable of achieving better video quality or lower bandwidth consumption.
3. The experimental section of the paper conducts comparisons with various methods on multiple datasets.

**Limitations:**

1. The main differences between QS-NeRV and other NeRV-based methods lie in the introduction of Invertible skip connections and the application of NeRV to SVC. However, the encoder-decoder structure and INN seem to be merely combined without innovative changes. Moreover, it is unclear why INN is only used for skip connections and not for $E^{L0}_{t}$. On the other hand, there have been many NN-based SVC methods (e.g., Swift[1]), and QS-NeRV does not exhibit much novelty in the network structure of EEN. Furthermore, the newly added InvNet and cascaded EEN lead to suboptimal inference speed. The fps of QS-NeRV-6M is 33, which can only be considered barely real-time.
2. Regarding the experiments, why does the evaluation on DAVIS include SSIM while Bunny and UVG do not? Apart from objective quality metrics such as PSNR and SSIM, are there any tests on subjective visual metrics? Moreover, in the original NeRV paper, the PSNR of NeRV, H264, and H265 on UVG is around 35 when BPP is 0.05, which is inconsistent with the data in Figure 8 of this paper. Additionally, QS-NeRV performs significantly worse than SHVC in Figure 8, making it seemingly difficult to use in practical video coding. Furthermore, can more details about the experiments be provided, such as the hardware used?
3. In terms of writing, Figure 1, Figure 2, and other figures take up too much space, resulting in relatively limited actual content in the paper.

[1]Dasari M, Kahatapitiya K, Das S R, et al. Swift: Adaptive video streaming with layered neural codecs[C]//19th USENIX Symposium on Networked Systems Design and Implementation (NSDI 22). 2022: 103-118.

**Suitability:**

3

---

### Official Review · Reviewer_ixDQ · 2024-05-10

**Rating:** 4
**Confidence:** 3

**Summary:**

This paper proposes a neural video representation method QS-NeRV to enable real-time quality-scalable decoding, which can well accommodate different quality requirements with the proposed Self-Learning Distribution Mapping Network (SDMN) and Extensible Enhancement Networks (EENs).

**Strengths:**

1. This paper is well-written and clearly organized.
2. The motivation is reasonable and the implementation is effective.
3. he experimental results can demonstrate the effectiveness of the proposed method.

**Limitations:**

1. In line 334 page 3, authors claimed “QS-NeRV achieves scalability by adjusting the number of EENs to accommodate diverse quality requirements.”. Herein, I would like to know why authors choose to adjust the number of EENs to control the quality rather than choose to achieve scalable feature representation like traditional methods. In addition, do authors think that regulating EENs is more like a post-processing solution, which can easily lead to a complexity increase?
2. The authors mentioned “train EEN in an overfitting way” many times in the manuscript. I think the authors should give a more reasonable and effective explanation instead of simply mentioning it.
3. The structure in Figure 4 is too difficult to understand, especially for “K, F, H, g”. It is hard to follow what this figure would like to express. How to understand “K, F, H, G are small DenseBlocks” in line 394 page 4 and why small DenseBlocks can be represented with different symbols.

**Suitability:**

2

---

### Official Review · Reviewer_bPnF · 2024-05-23

**Rating:** 2
**Confidence:** 4

**Summary:**

Authors propose QS-NeRV, an evolution of NeRV which exhibits scalable and real-time decoding capabilities on videos with various resolutions. The method is based on a hybrid architecture inspired by existing works such as HNeRV but uses stackable Enhancement Layers (EL) that improve the quality provided by Base Layers (BL), easing the encoding of a single video at various resolutions.

**Strengths:**

* The technical proposal is interesting and the paper is well-written.

* INR-based media compression is an interesting topic that has recently gained a lot of attention from the research community.

* The authors have the merit of addressing an issue loosely considered in the field, that is quality-variable
encoding.

**Limitations:**

* The UVG dataset comprises more than 7 videos, the same for DAVIS, but only a subset is considered. Which videos were extracted from the set and why were experiments not run on all the samples? Were those the same ones used by previous works? You need a strong reason to perform comparisons only on a portion of a dataset, otherwise, it may be considered cherry-picking the best results for your method.

* Why is HiNeRV not considered in most comparisons? It is only reported in Table 7 to demonstrate that the proposed QS-NeRV has better decoding times and HiNeRV cannot reach real-time decoding. Observing the official results (https://hmkx.github.io/hinerv/) HiNeRV outperforms DCVC, which means it could beat QS-NeRV as well in terms of quantitative metrics. These results should be included to specify that QS-NeRV does not beat every NeRV-based method in every aspect.

* Comparisons with recent COOL-CHIC methods could be included as well. Although those are not always referred to as INR-based, they are still based on overfitting and exhibit good performance.

* The concept of "real-time" presented in Section 4.5 is restrictive, as any speed which is greater than 30 FPS is considered to be real-time. This is true only if the original video is recorded/transcoded at 120 frames per second, which is not always true. For instance, UVG videos are available at 120 FPS. In addition, this evaluation is claimed to be conducted on the video containing 600 frames of 1920×960. Which video has been chosen and why? What's the original amount of frames per second? This is a limited experiment as one of the key points of the proposal is to enable real-time decoding, but a one-shot experiment on a single video is not enough to empirically prove the achievement.

* Referring to the study presented in Table 8, why does replacing invertible connections with direct
skip connections bring such an increase in the number of parameters?

* Regarding the study on joint vs separate training, it is not clear whether in joint training the whole model is trained for the same overall amount of epochs, which is 450 if I have understood correctly, or if the last phase where EEN is fine-tuned is simply omitted.

------------------
The experimental evaluation has some considerable limitations and important information is sometimes
missing in the discussion, plus some baselines are partially omitted. These shortcomings make the
conclusions unreliable.

**Suitability:**

2

---

### Meta-Review · Area_Chair_7WtF · 2024-07-08

**Recommendation:** Accept (Poster)
**Confidence:** 3

**Metareview:**

The paper received mixed reviews, tending more toward acceptance than rejection. The rebuttal helped to clarify a few things, and one reviewer changed the rating from borderline reject to borderline accept. Therefore, it can be considered for inclusion in the accepted pool of papers. However, the same reviewer still has reservations about its novelty, which could be improved when submitting the camera-ready version of the paper. Another reviewer still argues that "real-time is set based on a framerate of 30 FPS, which is unfair" -- I partially agree. For traditional video applications, this might be sufficient, but for higher framerates, this won't work anymore. It is recommended to explicitly address this to avoid confusion regarding the real-time aspect of this paper.